# Structural characterization of the microbial enzyme urocanate reductase mediating imidazole propionate production

Raminta Venskutonytė [1], Ara Koh[2,3], Olof Stenström[4], Muhammad Tanweer Khan[2], Annika Lundqvist[2], Mikael Akke [4], Fredrik Bäckhed [2,5,6] & Karin Lindkvist-Petersson [1,7✉]

The human microbiome can produce metabolites that modulate insulin signaling. Type 2 diabetes patients have increased circulating concentrations of the microbially produced histidine metabolite, imidazole propionate (ImP) and administration of ImP in mice resulted in impaired glucose tolerance. Interestingly, the fecal microbiota of the patients had increased capacity to produce ImP, which is mediated by the bacterial enzyme urocanate reductase (UrdA). Here, we describe the X-ray structures of the ligand-binding domains of UrdA in four different states, representing the structural transitions along the catalytic reaction pathway of this unexplored enzyme linked to disease in humans. The structures in combination with functional data provide key insights into the mechanism of action of UrdA that open new possibilities for drug development strategies targeting type 2 diabetes.

[1] Experimental Medical Science, Medical Structural Biology, BMC C13, Lund University, Lund, Sweden. [2] Department of Molecular and Clinical Medicine/ Wallenberg Laboratory, Institute of Medicine, University of Gothenburg, Gothenburg, Sweden. [3] Precision Medicine, Samsung Biomedical Research Institute, Samsung Medical Center, School of Medicine, Sungkyunkwan University (SKKU), Suwon, Republic of Korea. [4] Biophysical Chemistry, Center for Molecular Protein Science, Department of Chemistry, Lund University, Lund, Sweden. [5] Region Västra Götaland, Sahlgrenska University Hospital, Department of Clinical Physiology, Gothenburg, Sweden. [6] Novo Nordisk Foundation Center for Basic Metabolic Research, Section for Metabolic Receptology and Enteroendocrinology, Faculty of Health Sciences, University of Copenhagen, Copenhagen, Denmark. [7] LINXS - Lund Institute of Advanced Neutron and X-ray Science, Lund, Sweden. ✉email: karin.lindkvist@med.lu.se

Cardiometabolic diseases increase world-wide and recently an altered microbiome has been associated with type 2 diabetes[1,2]. Identification of microbial metabolites that contribute to type 2 diabetes have provided mechanistic understanding by which an altered microbiome may cause disease[3,4]. However, little is known about the enzymes producing such metabolites[3]. Interestingly, the fecal microbiota of type 2 diabetes patients has an increased capacity to produce imidazole propionate (ImP), which is catalyzed by the bacterial enzyme urocanate reductase (UrdA)[3]. UrdA from *Shewanella oneidensis* MR-1 was first characterized at the protein level by Bogachev and co-workers, revealing that it is in fact not a fumarate reductase, as was previously suggested, but a different enzyme[5]. UrdA provides an alternative pathway to metabolize the amino acid histidine by unidirectional reduction of urocanate to ImP[3,5] (Supplementary Fig. 1). It was predicted to be organized as a three-domain protein, with an N-terminal domain harboring a covalently bound FMN, an FAD-bound domain, and a mobile capping domain, the latter two forming a substrate-binding site[5]. To understand the structural organization and active site interactions, we determined X-ray structures of a two-domain construct of UrdA (hereafter UrdA′) and investigated the activity of the enzyme. The structure of UrdA′ was determined at four specific states: an ADP-bound structure without substrate at 1.1 Å resolution, and three FAD-bound structures in complex with either the substrate urocanate (1.56 Å) or the product ImP (1.40 Å), or without any ligand bound (2.56 Å). Both full-length UrdA and UrdA′ showed specific activity toward urocanate but not to fumarate, suggesting that the functional integrity and substrate specificity are conserved in the two-domain construct. The structural data provide detailed information on urocanate binding and insight into the structural changes upon conversion to ImP, and together with functional data allow us to identify key residues governing the substrate–enzyme interactions. Moreover, our detailed comparison of the four structures gives valuable insights into the mechanism of action of this enzyme, revealing that specific residues, in combination with larger conformational changes, regulate substrate access, and product release.

## Results

**X-ray structures of UrdA′ show details of substrate binding.** In this study, we used a variant of UrdA that excludes the N-terminal domain, because this variant does not require covalent flavinylation and can be expressed in *Escherichia coli* while still retaining the full substrate-binding site intact. Importantly, UrdA′ showed specific activity toward urocanate in the presence of FAD but not to fumarate (Fig. 1a, Table 1). To understand how the substrate and FAD cooperates, we crystallized UrdA′ in the presence of FAD and urocanate (hereafter called uro-FAD). The X-ray structure revealed that UrdA′ adopts a two-domain overall structure (FAD domain and clamp domain) connected by a hinge region (identified using the DynDom server[6]) (Fig. 1b). The data showed clear electron densities for both FAD and substrate

(Fig. 1c, Supplementary Fig. 2), where urocanate could be located between the FAD- and the clamp domains (Fig. 1c). The carboxyl group of urocanate forms an extensive interaction network, including a salt bridge to Arg560 and a hydrogen bond to backbone nitrogen of Gly563 on the FAD domain, as well as a hydrogen bond to Arg411 on the clamp domain and another hydrogen bond to His520 in the hinge region (Fig. 1d). All these contacts are conserved throughout the fumarate reductase (Frd) family in bacteria[7] as well as in yeast[8], and as revealed by our structural and functional data, essential for UrdA activity (Figs. 1d, 2a, c). Specifically, Arg401/402 in the well-studied soluble flavocytochrome c3 fumarate reductases (Fcc3) from *Shewanella spp.* (corresponding to Arg411 in UrdA) has been suggested to act as a proton donor in the conversion of fumarate to succinate[9,10]. In the uro-FAD structure, this arginine is similarly positioned 3.6 Å away from the C2 carbon of urocanate, suggesting that Arg411 acts as a proton donor in UrdA as well (Supplementary Fig. 3). The FAD is also positioned in a similar manner as in Frds (Supplementary Fig. 3), and accordingly the hydride transfer is likely to occur as in fumarate reductases, from nitrogen N5 of FAD[9]. In contrast to the conserved interactions of the carboxylate of urocanate in UrdA′, the molecular surroundings of the imidazole moiety of urocanate are different due to differences in the amino acid sequence as well as the conformation of the clamp domain. Specifically, the nitrogen atoms of the imidazole ring are positioned within hydrogen-bonding distance to oxygen O4 of FAD and to the side chain oxygens of Glu177 and Asp388 (Fig. 1d). Thus, depending on its tautomeric state, urocanate can act as a hydrogen bond donor to either of the Glu177 or Asp388 side chains (Fig. 1d). As the uro-FAD structure was obtained at neutral pH (UrdA is active at neutral pH[5]), the imidazole moiety of the urocanate is expected to be partially protonated. However, pH can vary in the intestine and this would affect the protonation state of the urocanate. When comparing sequences of UrdA from various bacteria from the human gut (Supplementary Fig. 4), Asp388 is conserved or exchanged for asparagine or glutamine, which all can form hydrogen-bonding networks as aspartate. On the other hand, Glu177 shows no conservation among the aligned sequences, suggesting that Asp388 is a critical residue for urocanate binding and UrdA activity, which was also confirmed by activity measurements (Fig. 2a, c).

Previously, we distinguished imidazole propionate-producing bacterial strains based on the presence of a tyrosine residue (Tyr373) in the hinge region of UrdA, in contrast to a histidine in fumarate reductases[3]. Interestingly, Tyr373 does not form a direct contact with urocanate in the structure (Fig. 1d), whereas the corresponding histidine in fumarate reductase from *Shewanella putrefaciens* MR-1 directly interacts with its substrate[10]. Based on this, we investigated if UrdA′Y373H had acquired fumarate reductase activity. In contrast to the absence of fumarate reductase activity in UrdA′wt we observed low fumarate reductase activity in UrdA′Y373H (Supplementary Fig. 5a), suggesting that the Tyr373 contributes to substrate specificity. To compare the activity of the two-domain UrdA′ with the full-length UrdA and investigate if the N-terminal domain, harboring a covalently bound FMN, might be critical for the specific activity, we have also produced a full-length UrdA (except the signal peptide) in *E. coli* and performed in vitro flavinylation for FMN attachment. The $K_m$ for the full-length enzyme was shown to be 0.43 mM (Fig. 2b, Table 1), which is similar to the value of 0.22 mM for the two-domain protein (Table 1). The different variants with substitutions in the active site of the full-length UrdA were also evaluated for enzyme activity. In agreement with the two-domain data, UrdA_R411A, UrdA_R560A, and UrdA_D388A showed no activity (Fig. 2c). Moreover, the full-length UrdA_Y373H also displayed

**Table 1 Enzyme kinetic parameters.**

| Enzyme | Substrate | $K_m$ (mM) | $V_{max}$ (µmol min$^{-1}$ mg$^{-1}$) | $K_{cat}$ (s$^{-1}$) |
|---|---|---|---|---|
| Two-domain WT | Urocanate | 0.22 ± 0.03 | 1.71 ± 0.12 | 1.43 |
| | Fumarate | n.d. | n.d. | n.d. |
| Full-length WT | Urocanate | 0.43 ± 0.03 | 5.57 ± 0.18 | 5.62 |
| | Fumarate | n.d. | n.d. | n.d. |

Data are presented as mean +/− SEM.

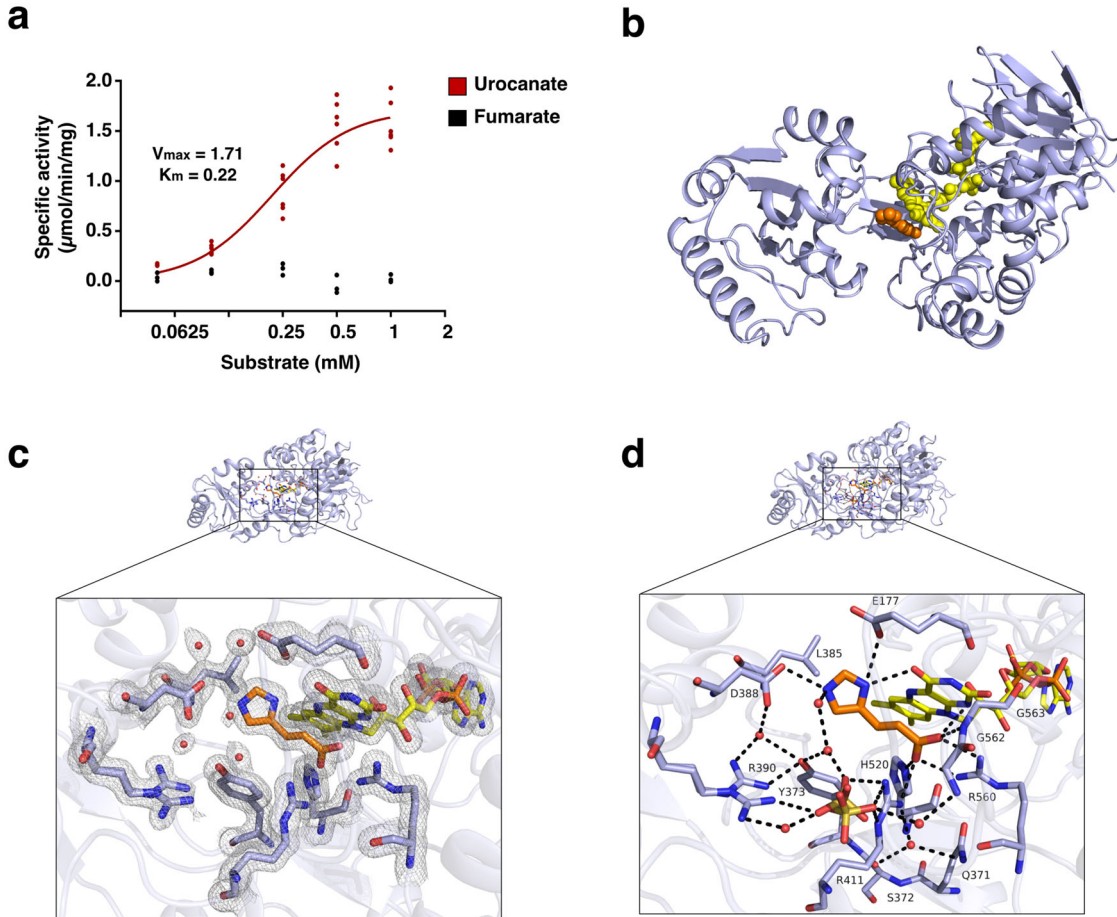

**Fig. 1 X-ray structure of UrdA′ in complex with urocanate and FAD and activity measurements. a** Activity and specificity of the two-domain UrdA′ toward varying concentrations of urocanate ($n = 6$ independent experiments) or fumarate ($n = 3$ independent experiments) measured in presence of FAD. **b** Cartoon representation of the three-dimensional structure of the two-domain UrdA′ with urocanate (orange) and FAD (yellow) shown as spheres. **c** Active site of uro-FAD structure, with urocanate (orange), FAD (yellow), active site residues (light blue), water molecules (red spheres), and 2mFo-DFc map contoured at 1σ and carved at 1.5 Å are shown. **d** Hydrogen-bonding network in the active site, urocanate (orange), FAD (yellow), active site residues (light blue), sulfate molecule in two alternative conformations and water molecules (red spheres) are shown. Potential H-bonds within 3.3 Å are displayed as dashed lines. Source data are provided as a Source data file.

### Table 2 Binding thermodynamics from ITC.

| Ligand | $K_d$ ($10^{-3}$ M) | $\Delta H^\circ$ (kJ/mol) | $-T\Delta S^\circ$ (kJ/mol) | $\Delta G^\circ$ (kJ/mol) |
|---|---|---|---|---|
| Urocanate | 0.28 ± 0.01 | 11.6 ± 0.2 | −31.2 ± 0.2 | −19.6 ± 0.1 |
| ImP | 1.08 ± 0.03 | 8.2 ± 0.2 | −24.5 ± 0.2 | −16.3 ± 0.1 |
| Urocanate–ImP | | 3.4 ± 0.3 | −6.7 ± 0.3 | −3.3 ± 0.1 |

Data are presented as fitted values +/− SD.

increased activity toward fumarate, thus further indicating the importance of His373 for ligand recognition in fumarate reductases (Supplementary Fig. 5b).

To further investigate the binding of urocanate and ImP to UrdA′, we determined the dissociation constants using isothermal titration calorimetry (ITC). The experiments show that ligand binding is endothermic and consequently entropically driven for both urocanate and ImP (Fig. 2d, Table 2). Based on the binding mode revealed by the X-ray structures with one ligand bound per protein molecule, we analyzed the ITC data with a fixed stoichiometry of 1:1. The binding affinity for urocanate is roughly a factor of 3–4 higher than that for ImP, as expected qualitatively for efficient enzyme activity. The difference in

dissociation constant is mainly due to a more favorable entropy of binding, $-T\Delta\Delta S^\circ = -6.7$ kJ/mol at 288 K, for urocanate over ImP, whereas the enthalpic contribution favors ImP binding, $\Delta\Delta H^\circ = 3.4$ kJ/mol (Table 2, Fig. 2d, Supplementary Fig. 6). We note that the fitted binding isotherm for urocanate appears to indicate a minor systematic deviation from the experimental data points that is not observed for ImP, possibly indicating that an additional process is involved in binding of urocanate. Based on the crystal structures, which reveal packing of two UrdA′ copies with an interface area of 880 Å² (calculated using the PISA[11] server), we speculate that this might entail a shift in monomer–dimer equilibrium of UrdA′. While the apparent dimer interface area is similar in the urocanate and ImP complex structures, the side chain of Arg410 shows flexibility and enters the dimer interface region in the ImP complex, suggesting that it might weaken the dimer in solution where additional crystal packing forces are not present. These observations led us to develop an additional model that accounts for both a protein monomer–dimer equilibrium and ligand binding to the monomer, which doubles the number of fitted parameters (Supplementary Note 1). However, fitting this model to the ITC data yields poor precision of the fitted parameters due to the decreased number of degrees of freedom. Thus, we opt to report only the results for the simpler model, while recognizing that the results

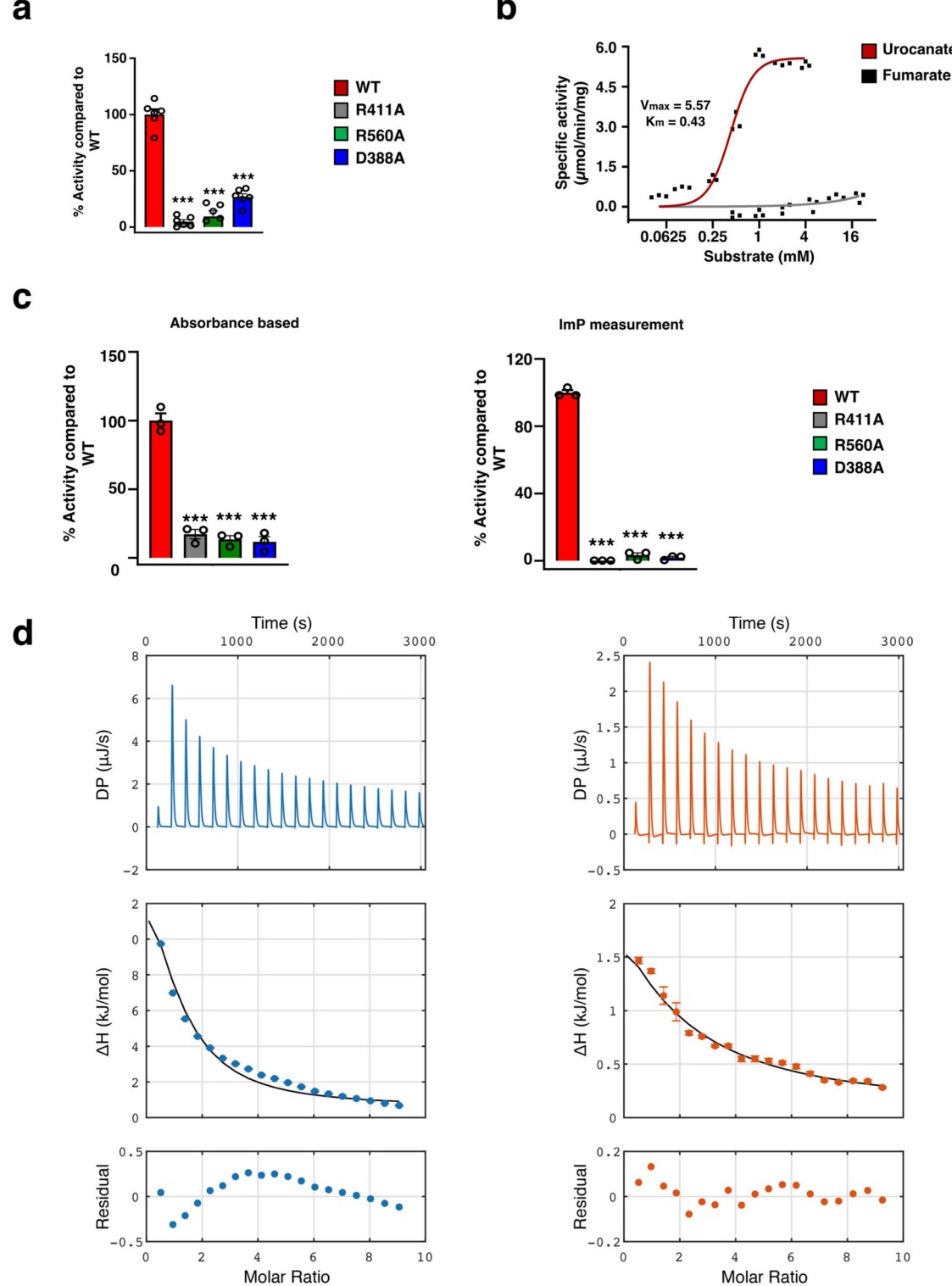

**Fig. 2 Activity measurements of UrdA. a** Relative activity of single substitutions, UrdA'$_{R411A}$, UrdA'$_{R560A}$, and UrdA'$_{D388A}$ compared to UrdA'$_{wt}$ ($n = 6$ independent experiments). Data are mean ± s.e.m. ***$P < 0.001$. One-way ANOVA with Dunnett's multiple comparisons test, $P$ value < 0.0001. **b** Activity and specificity of the full-length UrdA toward varying concentrations of urocanate ($n = 3$ independent experiments) or fumarate ($n = 3$ independent experiments) measured in presence of FAD. **c** Relative activity of single substitution variants, UrdA$_{R411A}$, UrdA$_{R560A}$, and UrdA$_{D388A}$ compared to the full-length UrdA$_{wt}$ in an absorbance-based measurement (left, $n = 3$ independent experiments) and in a direct measurement of ImP production (right, $n = 3$ independent experiments). Data are mean ± s.e.m. ***$P < 0.001$. One-way ANOVA with Dunnett's multiple comparisons test, $P$ value < 0.0001. **d** ITC measurement of urocanate (blue) and of ImP (orange) binding to UrdA', showing a representative dataset for each ligand. The full line shows the simultaneous fit to three replicate datasets. Errors in individual ITC data points are estimated based on the baseline uncertainty, as described[36]. Source data are provided as a Source data file.

for urocanate binding might represent effective parameters with underlying complexity.

**A large conformational change is observed upon ligand binding.** To investigate the overall catalysis mechanism of UrdA, from binding of urocanate and FAD to the changes accompanying the conversion to ImP, we have also determined the structures of apo UrdA′ and FAD (apo-FAD) and UrdA′ in complex with ImP and FAD (imp-FAD). In addition, we obtained a structure of UrdA′ with an ADP instead of FAD. Notably, this structure does not include any bound ligand, even though the sample was supplemented with urocanate (hereafter called apo-ADP). In the apo-ADP structure, an unambiguous density was present for ADP. In addition, density that could be best accommodated by a glycerol molecule was observed in the expected urocanate binding site (Supplementary Fig. 7). The closed conformation of the apo-ADP structure implies that the urocanate cannot be bound when ADP occupies the FAD-binding site, which was supported by the ADP-induced inhibition of UrdA′ activity (Supplementary Fig. 8). When comparing the apo-FAD structure (Supplementary Fig. 9) with apo-ADP structure, we also observed that the absence of the flavin group resulted in the clamp domain moving toward the FAD domain, thereby creating a closed, solvent inaccessible conformation. While with the FAD present, the clamp domain undergoes large structural rearrangements (~27° rotation) to create an open conformation (Fig. 3a). Moreover, the structures reveal that upon FAD binding, His521 adopts a different conformation to form a contact to the FAD O3 and His520 turns toward the binding site facing the flavin ring, possibly participating in a CH–π interaction (Fig. 3b). The different conformations seem to be stabilized by the loop 337–347 on the FAD domain, which is bending toward the binding site in the FAD-bound form, while in the ADP-bound form the loop is more open (Fig. 3b). The altered position of the 337–347 loop caused by FAD binding leads to opening of the clamp domain, allowing substrate access and subsequent enzyme–substrate complex formation. As seen in the uro-FAD structure, once urocanate enters the binding site, the clamp domain resumes a more closed state by interacting with the substrate (Fig. 3c, d), resulting in intermediate closure (22.5°) compared to a completely closed apo-ADP and largely open apo-FAD structures, and in this way stabilizes the active complex (Fig. 3c, d). It is worth mentioning that these three states of UrdA′ showing different extents of clamp-domain closure were crystallized in different space groups with different crystal contacts. This indicates that the domain opening is not restricted by crystal packing, but instead the conformation adopted by the protein in a ligand-bound versus ligand-free state affects the crystallization. When no ligand is bound, the flexibility of the clamp domain reduces the probability of obtaining well-diffracting crystals, as seen for apo-FAD, for which only lower quality crystals could be achieved, while the compact and closed apo-ADP and uro-FAD conformations resulted in a well-diffracting crystals. When comparing the UrdA′ structure to the Frd substrate–enzyme complex we see an important difference: UrdA′ in its substrate-bound form has a solvent-accessible channel, which is not present in Frds (Fig. 3e). Thus, while Frds utilize several residues to mediate proton transfer[7,9], a dependence on such pathway is not necessary in UrdA′, as water can directly access the substrate and provide protons.

**Arg411 adopts a different conformation in the presence of ImP.** In order to provide further insights into the catalytic mechanism, we also investigated the structure of UrdA′ in complex with ImP (imp-FAD) (Fig. 4a). Compared to urocanate, ImP adopts a slightly different conformation in the binding site,

as the absence of the double bond allows more flexibility of the molecule. The overall structure has the same conformation as uro-FAD, with root-mean-square deviation (rmsd) of ~0.3 Å (over 2912 atoms in Pymol). However, some key changes in the active site are observed. Strikingly, the suggested proton donor, Arg411, adopts a completely different conformation, facing away from the bound ligand (Fig. 4b, c). This change creates a space in the binding site, which might facilitate release of the product. In addition, the solvent-accessible channel to the substrate pocket becomes occluded (Supplementary Fig. 10a). This is partially caused by Phe243 turning toward and in between Glu177 and Asp388. Phe243 is now positioned in the same manner as in the apo-ADP structure. Moreover, Phe391 moves closer to the ligand and shows conformational flexibility, as judged from a less well-defined electron density modeled as multiple rotamers (Fig. 4c). Possibly, once Arg411 turns away from the bound product, the space created in the binding site allows slight movement of the clamp domain, bringing Phe391 closer to the binding site and resulting in an adjustment of Phe243. Overall, it seems that Phe243 and Phe391 represent a dynamic hotspot of the enzyme. Several datasets of both urocanate and ImP bound enzymes have been examined, and while the general trend is that Phe243 is in an open (with respect to the binding site) conformation in uro-FAD structures, and in a closed conformation in the imp-FAD structures, in some instances it can adopt an open conformation in the imp-FAD complex or a closed one in uro-FAD complex. Moreover, it also varies how well the electron density for Phe391 is defined in different datasets. These observations support our hypotheses that the flexibility of Phe243 and Phe391 might be important for the active site transition from the substrate-bound to the product-bound conformation. Moreover, Phe245 and Phe391 flank either sides of the aromatic ring of Phe243 and these three phenylalanines together cover the binding site (Supplementary Fig. 10b), forming a hydrophobic lid that could play a role in controlling water access to the channel, as well as substrate binding and product release. Taken together, we suggest that structural changes of Arg411, Phe243, and Phe391 initiate a conformation of the enzyme ready to release the product.

**The mechanism of action of UrdA.** Based on the structural and functional data presented, we suggest that apo-ADP represents an inactive state, while the binding of FAD induces movement of the clamp domain that enables binding of the substrate urocanate to form an active complex with a solvent-accessible channel for water-mediated proton delivery to Arg411, and subsequent protonation of the substrate. Once urocanate is reduced to ImP, Arg411 turns away accompanied by conformational changes, resulting in closure of the solvent-accessible channel by the hydrophobic lid and release of ImP, allowing for a new substrate molecule to enter the active site (Fig. 5). Interestingly, the ITC experiments showed that binding is entropically driven for both urocanate and ImP (Fig. 2d). This result might be explained in part by the difference between the apo-FAD and uro/imp-FAD structures. In the open conformation of apo-FAD the space between the clamp and FAD domains is filled by water molecules, whereas in the closed conformation of uro-FAD and imp-FAD the domain interface is significantly less solvent-exposed. Thus, ligand binding is very likely associated with desolvation of the interface between the two domains, which is expected to result in a favorable entropy of binding.

**Discussion**
The human intestine contains 100 trillion microorganisms producing diverse bioactive compounds[12]. As prerequisite to grow during anaerobic conditions in the intestine, where oxygen

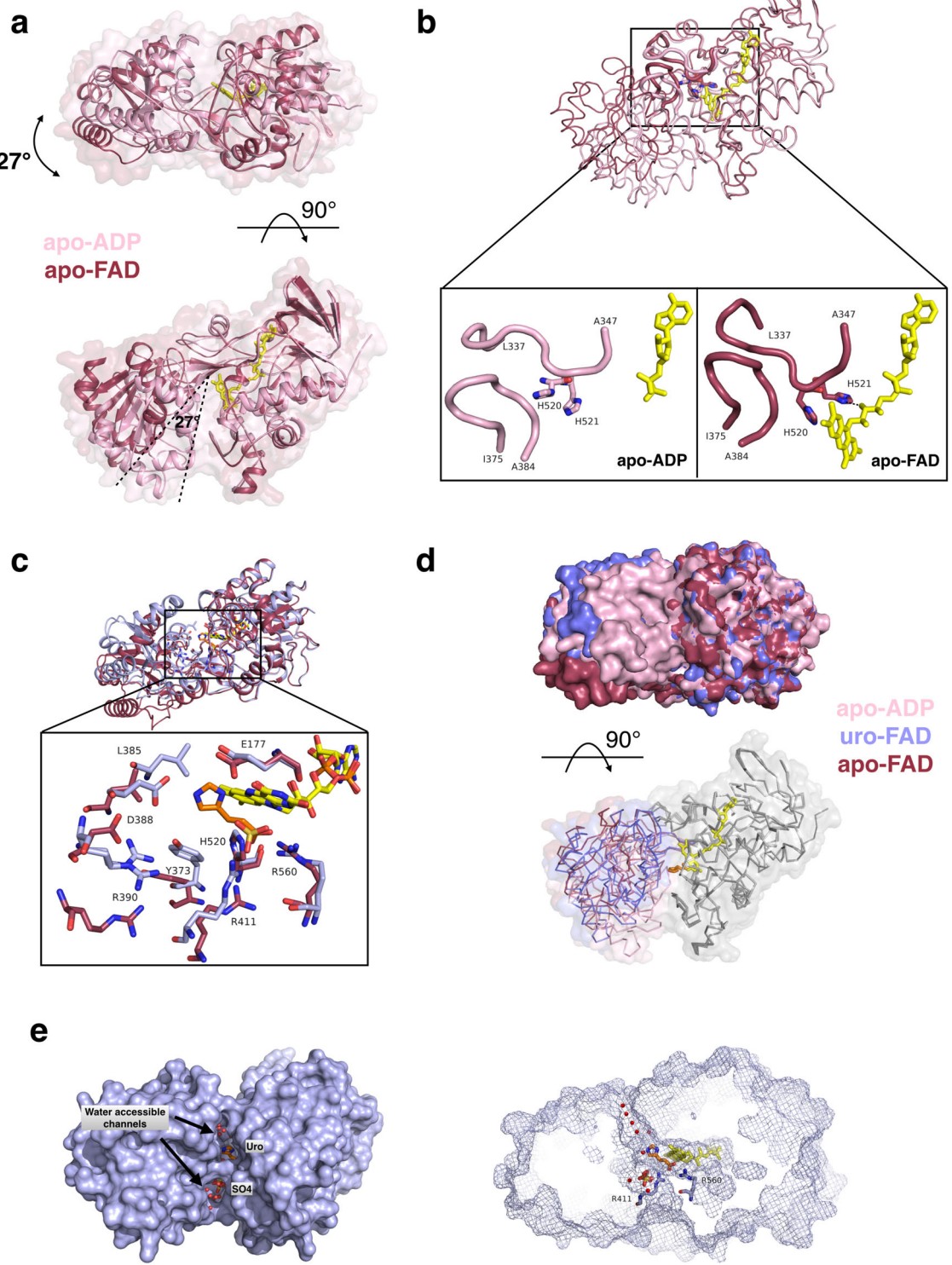

**Fig. 3 Conformational changes of the clamp domain upon ligand binding. a** Domain movement in closed/open structures. Superimposition of apo-FAD (red) and apo-ADP (pink) structures on FAD domain. The FAD and ADP are shown as sticks in yellow. Degree of clamp-domain rotation indicated as calculated using DynDom[6]. **b** FAD induced conformational changes. Comparison of apo-FAD (red) and apo-ADP (pink) structures, superimposed on the FAD domain. Flexible loops are shown in a thicker cartoon. His520 and His521 are shown as sticks, H-bond as a dashed line, ADP/FAD shown in yellow. **c** The clamp domain adopts different degrees of closure upon substrate binding. The shift in binding site residues upon urocanate binding compared to apo-FAD structure (superimposed on the FAD domain). The sulfate ion bound in apo-FAD structure in the same position as carboxy group of urocanate in the uro-FAD structure is shown sticks. **d** Clamp-domain closure comparison in apo-ADP (pink), apo-FAD (red), and uro-FAD (light blue) structures with urocanate (orange) and FAD (yellow). The structures were superimposed on FAD domain. **e** Water can access the active site in the substrate–enzyme complex. Surface representation of uro-FAD (left), mesh representation of the surface at the angle to illustrate water access (right). Urocanate (orange), FAD (yellow), water molecules (red spheres), sulfate, Arg411, and R560 (light blue) are shown.

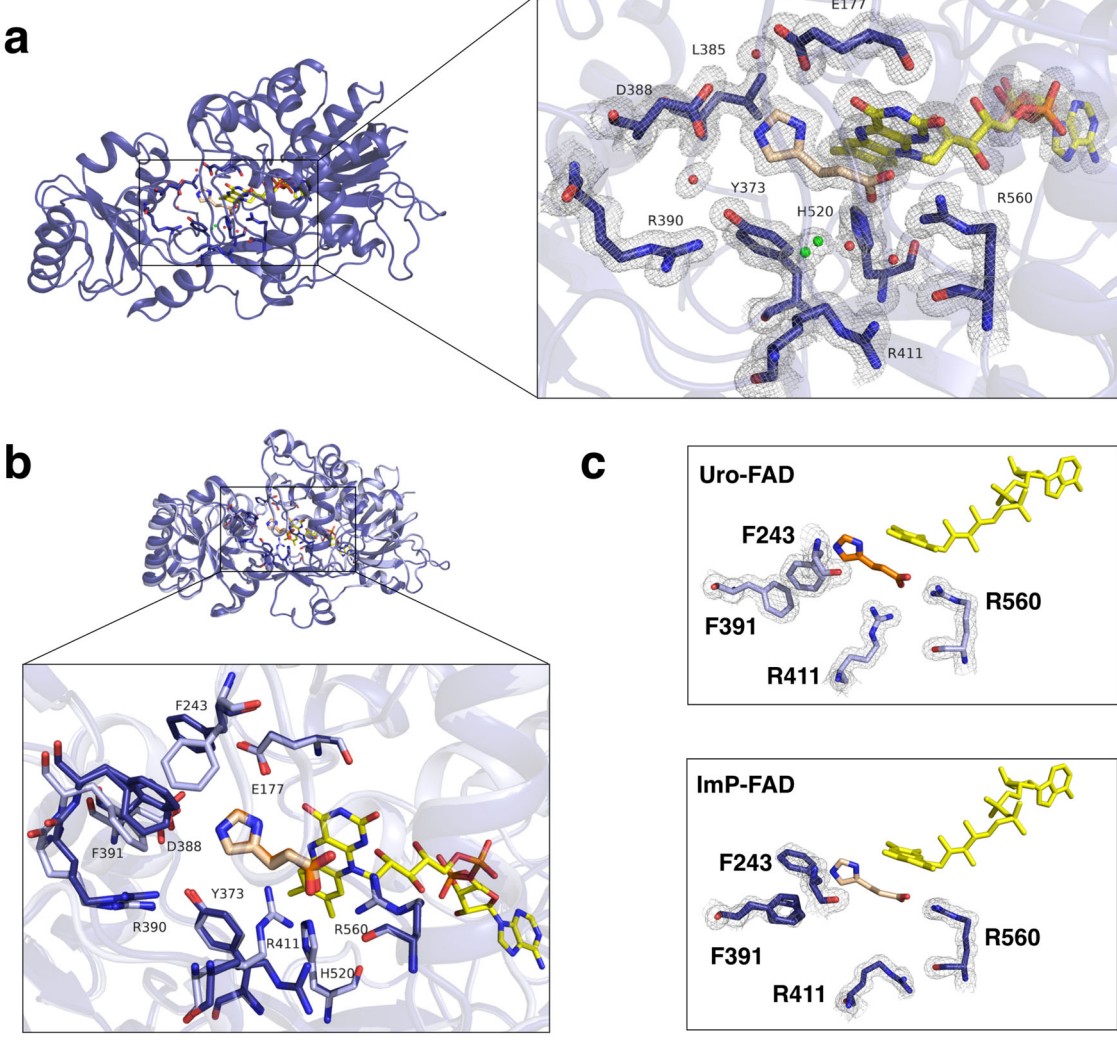

**Fig. 4 X-ray structure of UrdA′ in complex with imidazole propionate and FAD. a** Active site in the imp-FAD structure, ImP (wheat), FAD (yellow), active site residues (blue sticks), water molecules (red spheres), and 2mFo-DFc map contoured at 1σ and carved at 1.5 Å are shown. **b** Comparison between uro-FAD (light blue) and imp-FAD (blue) active site. The two structures were superimposed on the FAD domain. Urocanate (orange) and imidazole propionate (wheat) are shown. **c** Different conformations of Phe243, Phe391, and Arg411 in substrate/product-bound states. 2mFo-DFc map contoured at 1σ is shown for these residues.

cannot be used as an electron acceptor, alternative ways to dispose electrons have evolved. UrdA provides one such example by reducing urocanate to ImP, which has recently been related to the pathogenesis of type 2 diabetes[3]. Thus, an improved understanding of the unexplored enzyme UrdA is of clear medical interest. Here we examined UrdA by combining structural and functional analyses. We determined the structure of UrdA′, comprised of the two domains forming the active site, in four different states and confirmed its activity and specificity for urocanate by activity assays.

The overall structural properties of UrdA′ are similar to the well-studied soluble flavocytochrome c3 fumarate reductase from *Shewanella oneidensis*, but specific residues contribute to selectivity for urocanate in UrdA. In Fcc3, there are six residues that are critical for substrate binding and fumarate reduction[10,13], and the corresponding residues of the soluble yeast fumarate reductases have similar function[8]. The residues of the FAD domain, interacting with the carboxyl group of fumarate/urocanate, are sequentially and structurally conserved between UrdA and fumarate reductases (Supplementary Fig. 3). In contrast, the histidine (His364/365 in Fcc3) that make a bond to the C1

carboxylate of fumarate corresponds to a tyrosine in UrdA. This histidine is known to be important for fumarate binding, but not for the catalysis[13]. Urocanate has an imidazole moiety instead of carboxylate and thus UrdA does not require a histidine to be present in this position for ligand recognition. In fumarate reductases, Arg380/381, Glu377/378, and Arg401/402 form a proton transfer pathway to the substrate[7,10]. While both arginines also are found in UrdA (Arg390 and Arg411), Glu377 is not conserved and the described pathway is not present (Supplementary Fig. 3). This supports the hypothesis that UrdA does not require specific residues for proton transfer as there is a solvent-accessible channel to the active site in the substrate-bound state. Furthermore, UrdA′ was shown to gain activity for fumarate upon replacement of Tyr373 with histidine, confirming that this residue contributes to substrate specificity. Nevertheless, as only low activity was observed, this result also suggests that there must be other parts of the protein that contribute to substrate selectivity. When superimposing the structure of fumarate reductase, Fcc3 with UrdA, the region that differs the most is a helix (aa 383–391 in UrdA and 373–380 in Fcc3) on the mobile domain, which is structurally close to the imidazole ring of urocanate. This

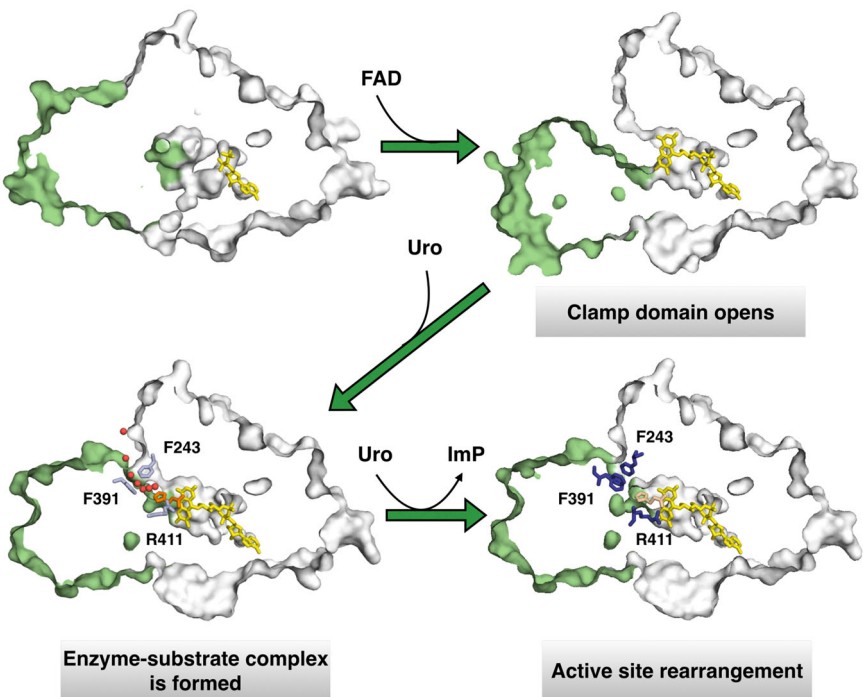

**Fig. 5 The structural mechanism of action of UrdA.** Schematic illustration of the conformational changes upon FAD and substrate/product binding in the active site. Clamp domain shown in green and FAD domain in white, ADP/FAD (yellow), urocanate (orange), imidazole propionate (wheat), and water molecules (red spheres) indicated.

helix in UrdA includes both Asp388, which interacts with the substrate, and part of the hydrophobic lid consisting of Phe391, Phe243, and Phe245. The hydrophobic lid displays flexibility, which might be a contributing factor in the release mechanism of ImP. Interestingly, the role of aromatic residues regulating product release has been described for another flavoenzyme, aryl-alcohol oxidase, where two phenylalanines and one tyrosine residue form an aromatic triad over its active site, separating it from the outer environment. Particularly, one of these phenyla-lanines was shown to contribute to substrate access and product release, as well as prevention of oxygen diffusion into the binding site[14,15]. In fumarate reductases, these phenylalanines are not conserved and the clamp domain is closer to the FAD domain, bringing the mentioned helix toward the binding site. Such clo-sure is not possible in UrdA′, as it would result in a steric clash with the ligand. The shift in the helix also explains the require-ment for Fcc3 to have Glu377/378 and Arg380/381, which par-ticipate in proton transfer, aligned with Arg401/402 (Arg411 in UrdA). Interestingly, when further examining the overlaid Fcc3 and UrdA′ structures, the carboxylate of Glu377 in Fcc3 superimposes exactly with a sulfate molecule found in the uro-FAD structure (Supplementary Fig. 11). This sulfate seems to mimic the carboxyl group of Glu377 in Fcc3 by bridging between Arg390 and Arg411. In the imp-FAD structure, which was obtained without sulfate ions present, there is an extra density in this position that could be best accommodated by a glycerol molecule and a chloride ion. It is not clear if this has any bio-logical relevance; rather it is likely a crystallization artefact that serves to stabilize the protein conformation in the crystal. Fur-thermore, when comparing UrdA with other oxidoreductases, three amino acid substitutions stand out. In addition to the tyr-osine versus histidine at position 373, Phe391 of UrdA corre-sponds to a conserved glycine in other enzymes; and Glu377, which is part of the proton transfer pathway in Frds, corresponds to an alanine in UrdA. Together these three residues distinguish

the substrate-binding site of UrdA from other related oxidor-eductases (Supplementary Fig. 12).

In contrast to the rather well-conserved C-terminal ligand-binding domains of the oxidoreductases, the N-terminal domain varies both in sequence and function, examples being the heme-bound cytochrome domain in soluble Fcc3[7] and the iron-sulfur domain in quinol Frds[16], while yeast soluble fumarate reductase[8] and L-aspartate oxidases[17] lack the N-terminal domain. These differences reflect the variations in the electron transfer chain of different enzymes. UrdA has an N-terminal domain with a cova-lently bound FMN cofactor. FMN-containing domains are also found in NADH:quinone oxidoreductase[18] and in the recently described extracellular fumarate reductase from Gram-positive bacteria[19]. The latter study suggested that the N-terminal domain, where FMN also is covalently attached, directs the electron transport to the protein[19]. The fact that the periplasmic UrdA also has such a motif might indicate a similar way of acquiring electrons, possibly from the membrane-bound quinone via cyto-chrome c or a different protein, but this area is open for further investigations. Different oxidoreductases seem to have evolved conserved C-terminal ligand-binding domains while varying the electron-accepting module to adapt to different cellular environ-ments. Conserved ligand-binding domains are expected in enzymes that utilize the same or similar substrates, like dicarboxylate in fumarate reductase or L-aspartate oxidase. However, urocanate is significantly different from the substrates of other oxidoreductases, which is reflected by the large structural changes needed to accommodate the imidazole moiety. Interestingly, a structural similarity search for proteins resembling UrdA′ (DALI server)[20], yielded a high score for 3-ketosteroid delta 1-dehydrogenase in addition to the soluble fumarate reductases. The former serves as an example of an oxidoreductase with conserved structural organiza-tion, especially the FAD domain, but with a very different sub-strate[21]. This example together with UrdA illustrates how the local environment of the binding site can be adapted to a particular

ligand, while keeping the overall structural organization homologous.

While the extended biology of full-length UrdA is an exciting area for the future studies, in this study we examine the catalytic mechanism of the enzyme using structural and functional data of substrate-binding domains of UrdA. We demonstrate that the two-domain UrdA′ is an excellent system to study ligand–protein interactions. The possibility to achieve high yields of protein and high-quality crystals allows us to obtain detailed information on bound molecules in the active site and provides a useful tool for structure-based drug design for designing inhibitors for future treatment of type 2 diabetes. In conclusion, here we report the first high-resolution structures of UrdA, an enzyme able to convert urocanate, a metabolite of L-histidine, to the non-metabolizable product imidazole propionate, which is known to interfere with human glucose metabolism. Key insights into the conformational changes associated with substrate and product binding suggest strategies for future intervention targeting the active site.

## Methods

**Two-domain UrdA′ expression and purification**. The expression construct comprising two domains, the clamp and the FAD-binding domain, of the uroca-nate reductase from *Shewanella oneidensis* (strain MR-1) consisted of amino acids 130–582 as well as a 6xHis tag at the C-terminus. The synthetic gene in vector pET-24a (+) was transformed into *E. coli* TUNER (DE3) strain and expressed using LB media supplemented with 50 μg/ml kanamycin in shaker flasks at 37 °C, 150 rpm until the cell density reached the OD of 0.5–0.8. The cultures were cooled on ice and induced with 0.5 mM IPTG. The expression was continued overnight after induction at 20 °C. The cells were collected by centrifugation and stored at −80 °C. For the purification, the cell pellet was resuspended in the wash buffer (50 mM sodium phosphate pH 8.0, 300 mM NaCl, 20 mM imidazole) and disrupted by sonication. After clearing the lysate by centrifugation, it was loaded onto a HisTrap HP 5 mL column (GE Healthcare). The protein was eluted by imidazole gradient. The peak fractions had a bright yellow color, as expected due to bound FAD. The protein eluted in one large asymmetrical peak, corresponding to a monomer and a dimer. The main peak fraction consisting of mostly monomeric UrdA′ was used for further experiments. The substituted variants of UrdA′ were produced essentially in the same way as the wild-type UrdA′, but a BL21 (DE3) *E. coli* strain was used and instead of binding to an affinity column, the protein was captured by NiNTA affinity resin (Qiagen) and eluted in batch.

**Full-length UrdA expression and purification**. The full-length UrdA corresponds to the full sequence of the *Shewanella oneidensis* (strain MR-1) except for the first 25 amino acids comprising the signal peptide. Expression and purification were performed in a same way as described for the two-domain UrdA. After purification, the protein was subjected to in vitro flavinylation to obtain the covalently attached FMN at the N-terminal domain. The in vitro flavinylation was performed at 8 °C overnight using ApbE enzyme in the buffer based on previously reported conditions[22]. The ApbE expression vector was prepared using a synthesized gene (ThermoFisher) for ApbE (Uniprot A5F5Y3) excluding the signal peptide and with a 6 histidine tag at the C-terminus, which was then cloned into a pBADM-11 vector, with a deleted N-terminal part coding for a HisTag and a cleavage site, using standard methods. The ApbE was expressed in *E. coli* using L-arabinose for the induction. The protein purification was performed in the same way as for the UrdA. After affinity and size-exclusion chromatography SDS-PAGE analysis indicated that the ApbE is only partially pure. Each batch of the partially purified enzyme was tested in a small scale to estimate the amount of the sample needed for UrdA flavinylation. The flavinylated UrdA was checked on the SDS-PAGE under the UV light prior to the staining of the gel (Supplementary Fig. 13). After flavi-nylation, the UrdA-ApbE mixture was subjected to size-exclusion chromatography and the flavinylated UrdA collected for enzyme assays.

**Site-directed mutagenesis**. The single mutants of UrdA′ and UrdA (D388A, R411A, R560A, and Y373H) were prepared using QuikChange® site-directed mutagenesis kit (Agilent) or Phusion® polymerase following standard protocols. The mutations were confirmed by sequencing (Eurofins, Germany or Macrogen Europe, The Netherlands).

**Crystallization**. Before crystallization the proteins were purified by size-exclusion chromatography using Superdex 75 or Superdex 200 column (GE Healthcare) in the buffer consisting of 20 mM HEPES pH 7.0, 150 mM NaCl, 10% glycerol, 0.5 mM TCEP, and peak fractions collected and concentrated. Hanging drop vapor diffusion method at 18 °C was used for crystallization and the initial conditions were found using a Hampton crystallization screen. For the ADP-bound structure, the crystals were obtained by mixing 1 μL of ~30 mg/mL protein solution

supplemented with 2.9 mM urocanic acid (Sigma 859796) with 1 μL of reservoir solution consisting of 20% PEG4000, 0.1 M MgCl₂, and 0.1 M HEPES pH 7.0. No additional FAD was added, as the enzyme is expected to acquire the cofactor during the recombinant expression in *E. coli* and as observed by the yellow color of the purified protein sample. For the urocanate and FAD-bound structure the crystals were obtained using protein solution of ~30 mg/mL, supplemented with 3.4 mM FAD and 3.0 mM urocanic acid and a reservoir solution consisting of 27% PEG8000, 0.3 M (NH₄)₂SO₄, 0.1 M HEPES pH 7.0. Crystals in complex with imidazole propionate were obtained using ~26 mg/mL protein sample in 20 mM HEPES pH 7.0, 150 mM NaCl, supplemented with 3.2 mM FAD and 7 mM imi-dazole propionate (Sigma 77951) in a condition consisting of 20% PEG8000, 0.2 M NH₄Cl, and 0.1 M HEPES pH 7.0. Crystals for apo-FAD-bound structure were obtained using a protein solution of ~30 mg/mL in 20 mM HEPES pH 7.0, 150 mM NaCl buffer. Prior crystallization protein sample was supplemented with 3.2 mM FAD. Crystal used for data collection was obtained by mixing 1 μL of protein solution with 1 μL of 0.1 M Tris pH 8.5, 2 M (NH₄)₂SO₄ reservoir solution, and 0.2 μL of 30% xylitol from the additive screen (Hampton, USA). For the data collection, the crystals were flash-frozen in liquid nitrogen after immersing into cryo-protectant consisting of the reservoir solution with addition of 20% glycerol.

**Data collection and structure determination**. The X-ray data for the ADP-bound structure was collected at PETRA III P13 beamline at EMBL in Hamburg, Germany. Three datasets from two crystals from the same crystallization drop were used for structure solution. Two of the datasets were obtained using helical data collection on the same crystal. Each dataset was processed in XDS[23] and then merged and scaled in AIMLESS[24] within the CCP4 program suite[25]. The structure was solved by molecular replacement using Phaser[26] within Phenix[27]. The search model was modified fumarate reductase (pdb id 1D4C[10], chain A), which was modified by removing residues 1–122 (HEME domain, not present in UrdA′ construct) and HET atoms. Data for the uro-FAD, imp-FAD, and apo-FAD structures were collected at Paul Scherrer Institute, Villigen, Switzerland at the beamline X06SA. Datasets from single crystals were processed in XDS[23] and CCP4[25]. The uro-FAD and apo-FAD structures were solved by MR in Phaser[26] using two ensembles—the clamp domain and the FAD domain of the apo-ADP structure and the imp-FAD structure was solved using the uro-FAD structure as a search model. The structures were solved in P 2₁, P 3₁ 2 1, and P 6 2 2 space groups for apo-ADP, uro/imp-FAD, and apo-FAD structures, respectively. The resulting models were further subjected for structure building in AutoBuild[28] within Phenix[27]. Further model editing and refinement were performed in Coot[29] and phenix.refine[30]. Anisotropic B-factors and riding hydrogens were used in apo-ADP and uro/imp-FAD structures, while the apo-FAD structure was refined with isotropic B-factors and TLS. All residues except the last histidine residue of the 6xHis tag in apo-ADP and all the 6 residues of the His-tag in the three other structures, were modeled. In the uro-FAD and imp-FAD positive density at the C-terminal was observed, however, it was not clear how to model the His-tag. We have collected and examined several datasets for each structure, especially for uro-FAD and imp-FAD. Maps clearly indicated a sulfate ion present in two alternative conformations in the vicinity of the substrate (close to Arg411) in uro-FAD structures, which could be unambiguously modeled. However, in the imp-FAD structure, we chose to use the data from the crystal without sulfate, to avoid extra interacting molecules in the binding site and to be able to more clearly evaluate the state of the enzyme, as movement of Arg411 created space for extra water molecules and possible alternative positions for sulfate ions. Nevertheless, in imp-FAD structure crystallized without sulfate ions, a glycerol molecule (glycerol was used as a cryo-protectant while freezing the crystals) and putative chloride ions were found instead in a similar position. In the apo-FAD structure, a sulfate ion was bound in the active site as well, interacting with Arg560, while the apo-ADP had a glycerol and two chloride ions in the binding site. These extra ligands in the binding site might act as stabilizing molecules for the crystallization. In the high-resolution uro-FAD and imp-FAD structures, the majority of the binding site residues as well as substrate/product were very well-defined. However, some positive and negative Fo-Fc peaks were also present (Supplementary Fig. 14). Glu177 residue presented extra positive density and trials to model an alternative conformation were not convincing of a geometrically correct and fitting alternative rotamer. Thus, it might be an indication of flexibility of the Glu177. Other positive peaks could of course represent noise as well as dynamic water molecules. FAD could be unambiguously modeled in a clearly defined electron density. Although the crystals had a yellow color, it is hard to judge what redox state of the FAD was present during data collection, as X-ray exposure can cause the reduction of the flavin[31]. The studies on free flavins showed that the isoalloxazine ring is expected to be in a planar conformation when oxidized and becomes bent upon reduction, however, it was also shown that the conformation of the isoalloxazine ring largely depends on the environment[32]. For example, the distortion of the flavin was shown to be induced by the ligand binding in cholesterol oxidase[33]. Moreover, it has been shown that the conformation of the flavins can influence their redox potential[34]. In our high-resolution structures of uro-FAD and imp-FAD, the flavin ring is not completely planar and slight distortion was observed. It is possible that it has become partially reduced, however, it might be in a more reactive distorted oxidized state. Imidazole propionate ligand was drawn using MarvinSketch (version 19.12 Che-mAxon) and a SMILES string obtained used in elbow[35] within Phenix[27] to generate a cif ligand file and the urocanate ligand was retrieved from the CCP4 monomer library. The structure figures were prepared in Pymol (Schrödinger). The domain movement

was calculated using DynDom[6], comparing the different structures. The crystallographic data and refinement statistics are provided in Supplementary Table 1. The structures were deposited to the Protein Data Bank with the following IDs: 6T85 for apo-ADP, 6T86 for apo-FAD, 6T87 for uro-FAD, and 6T88 for imp-FAD.

**Activity assay**. For UrdA activity assay, methyl viologen reduced with sodium dithionite in 100 mM Tris-HCl (pH 8.0) was used as an electron donor. Each enzymatic reaction was performed under anaerobic condition in 200 μL buffer (50 mM HEPES/Tris (pH 7.0) and 100 mM NaCl) using 500 ng of UrdA′, FAD (0.03 mM at final concentration), reduced methyl viologen (0.68 mM at final concentration) and varying concentrations of urocanate or fumarate. The reductase activity was determined at different time points (0, 2, 5, 6, 8, 10, and 12 min) by subtracting absorbance ($OD_{620}$) from controls without substrate or enzyme, which is dependent on oxidation of the reduced methyl viologen. To calculate the initial reaction velocity, Beer–Lambert equation was used ($A$ = delta absorbance at 620 nm/min, $\varepsilon$ = 13.7/mM·cm, and path length = 0.56 cm). For relative activity (% activity) comparisons, $OD_{620}$ subtracted from controls (without substrate) at 5 min reaction time was used. Using the reaction volume of 0.2 mL, the velocity was calculated as μmol × min$^{-1}$ mg × protein$^{-1}$. Kinetic parameters were calculated by nonlinear regression analysis using GraphPad Prism Software. For the WT enzyme the better fit was obtained using the sigmoidal allosteric equation (Hill coefficient – 2). We cannot fully explain such result, however, we believe this might be due to the complexity of the enzyme–substrate interaction, particularly a shift in monomer–dimer equilibrium of the protein. This was also indicated by the ITC measurements as discussed in the "Results" section.

To directly measure the activity, ImP was measured as follows. The samples were extracted using 3 volumes of ice-cold acetonitrile and internal standards (imidazole propionate-$^{13}C_3$) were added before drying the samples under a flow of nitrogen. Then, they were reconstituted with 5% HCl (37%) in 1-butanol, subjected to n-butyl ester derivatization, and finally reconstituted in water:acetonitrile (9:1). Samples (5 μl) were injected onto a C18 BEH column (2.1 × 100 mm with 1.7 μm particles; Waters, Milford, MA) and separated using a gradient consisting of water with 0.1% formic acid (A-phase) and acetonitrile with 0.1% formic acid (B-phase). ImP was detected by UHPLC-MS/MS analysis using multiple reaction monitoring of the transition 197/81.

**Isothermal titration calorimetry**. Isothermal titration calorimetry (ITC) experiments were performed on a MicroCal PEAQ-ITC instrument (Malvern Panalytical) at a temperature of 288 K and a stirring speed of 500 rpm. Experiments were performed by titrating 10 or 12 mM of urocanate or ImP, respectively, into the cell containing UrdA′ at a concentration of 0.235 mM. Three replicate experiments were performed for each complex. Peak integration was performed using NITPIC (version 1.2.2)[36]. A single-site binding model was fitted simultaneously to each replicate with the stoichiometry fixed to 1:1, using in-house MATLAB (R2017b) routines employing error estimation based on 500 Monte Carlo data points[37]. The heat released or absorbed during the $i$th injection is given by

$$\Delta Q_i = Q_i - Q_{i-1} + (V_i/V_0)[Q_i - Q_{i-1}]/2 + Q_{\text{off}} \quad (1)$$

where $V_i$ is the volume of the ith injection, $V_0$ is the cell volume, $Q_{\text{off}}$ is an offset parameter that accounts for heats of mixing, and $Q_i$ is the heat function following the ith injection:

$$Q_i = (\Delta H V_0/2)\left[\alpha - \sqrt{\alpha^2 - 4nM_iX_i}\right] \quad (2)$$

where $\alpha = nM_i + X_i + K_d$, $n = 1$, and $M_i$ and $X_i$ are the total concentrations of protein and ligand, respectively, in the cell at any given point of the titration. The free energy and entropy of binding were subsequently determined using the relationships $\Delta G° = RT \ln(K_d)$ and $-T\Delta S° = \Delta G° - \Delta H°$. In addition, a higher-order model that includes dimerization of the free protein was tested. The detailed expression for this model is given in the Supplementary Information.

**Reporting summary**. Further information on research design is available in the Nature Research Reporting Summary linked to this article.

## Data availability
Atomic coordinates for the crystal structures have been deposited in the Protein Data Bank under accession numbers: 6T85 for apo-ADP, 6T86 for apo-FAD, 6T87 for uro-FAD, and 6T88 for imp-FAD. Materials are available upon reasonable request. Source data are provided with this paper.

## Code availability
MATLAB scripts for fitting ITC data are available on GitHub (https://github.com/OlofStenstrom/ITCdimer).

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

## Acknowledgements

We thank synchrotron facilities at DESY, Hamburg (EMBL), PSI, Villigen, and MAXIV, Lund for providing beamtime and assistance. The members of Dr. Pontus Gourdon lab at the University of Copenhagen are thanked for data collection at PSI. Lund Protein Production Platform is acknowledged for assistance in initial crystallization and protein production experiments, Katharina Beck for helpful suggestions, and Pål Stenmark for critical reading of the manuscript. This study was supported by European Research Council (ERC) Proof of concept (780659), and Swedish Research Council (2016-01319 to K.L.-P., 2017-05816 to K.L.-P., 2018-4995 to M.A.). F.B. is a recipient of an ERC Consolidator Grant (615362; METABASE) and is the Torsten Söderberg Professor in Medicine.

## Author contributions

Designed the experiments: R.V. and K.L.-P. (structural biology), A.K., M.T.K., and F.B. (enzyme activity assays), O.S. and M.A. (ITC). Conducted the experiments: R.V. (structural biology), A.K. and A.L. (enzyme activity assays), O.S. (ITC). Analyzed data: R.V. and K.L.-P. (structural biology), A.K. and F.B. (enzyme activity assays), O.S. and M.A. (ITC). R.V., A.K., F.B., and K.L.-P. wrote the manuscript with contributions from all authors.

## Funding

## Competing interests

A.K., M.T.K., F.B., and K.L.-P. are shareholders of Implexion Pharma. R.V., A.L., O.S., and M.A. declare no competing interests.
