## [Peer Review File · Nature Communications]

Reviewers' comments:

Reviewer #1 (Remarks to the Author):

With the discovery of the multiple roles and the diversity of the microbiome in our intestine, there is literally an explosion of studies on the association between “endogenous” microbial metabolites and disease. Imidazolepropionate has been associated to type 2 diabetes and impaired glucose tolerance. This compound is generated by a reductase in the microbial pathway for histidine catabolism. The imidazolepropionate-producing urocanate reductase belongs to the well-known family of the succinate dehydrogenases/fumarate reductases. These enzymes catalyze the oxidation and/or reduction of dicarboxylic substrates and there is wealth of knowledge about them.

In this manuscript, Venskutonytė et al describe their excellent crystallographic work revealing the three-dimensional structure of the urocanate reductase. High-resolution structures are described for the substrate and product complexes of the holoenzyme (FAD-bound) as well as for the enzyme in the absence of substrate/product and flavin. Mutagenesis data nicely support the conclusions drawn from the structural analysis.

Urocanate reductase displays all the typical features of this enzyme family: an active-site arginine that function as proton donor and acceptor, a dynamic capping domain that opens and closes the catalytic site, a pair of Arg side chains that anchor the substrate carboxylate group, and a substrate-binding mode whereby an hydride anion can be transferred from/to the flavin. The characteristic features of urocanate reductase are mostly associated to the substrate specificity: the substrate site contains a niche for the binding of the substrate imidazole side chain. It is noticeable that the flavin itself is an H-bonding partner for the imidazole.

Technically, the manuscript is of the highest quality. The pictures with the electron densities speak by themselves. The manuscript is instead somewhat less strong with regard to the biological insight. There is no discussion about how the structural knowledge could be used for drug inhibitor design. There is very little discussion about enzyme evolution in relation to the fumarate activity reductase activity displayed by an active-site mutant. E.g. are there enzymes of the same class that operate on bulky side-chain substrates? Likewise, there is little discussion about the mechanistic implications. What is the pH optimum for the enzyme? Is there a catalytically optimal protonation state for the substrate imidazole? Is the enzyme confirmed to be unidirectional (reference 5) or can it catalyze also imidazolepropionate oxidation? How does the enzyme position itself on the membrane surface? Where is the active site entrance (Figure 2E) located with respect to the membrane surface? Is the reductant (a quinone? This is not at all discussed in the manuscript) derived from the membrane? If so, how does it bind to the enzyme?

Minor point

Lines 102-107 of page 5: please clarify the text on the Y373 mutants with an explicit indication of their steady-state parameters k_{cat} and K_m .

Reviewer #2 (Remarks to the Author):

The paper by Venskutonyte et al. presents the 3D structure of urocanate reductase (UrdA) from bacterium *Shewanella oneidensis*. Type 2 diabetes is associated with altered gut microbiome. The authors showed previously that this microbiome alteration results in an increase in imidazole propionate produced by UrdA-containing bacteria. Importantly, they revealed that administration of imidazole propionate in mice results in an impaired glucose tolerance (Koh et al., 2018). However, it remained unknown, whether this enrichment by UrdA-containing bacteria is the etiology or the consequence of diabetes. To answer this question, it is necessary to design specific inhibitors for UrdA, which was hampered by the lack of 3D structure of this enzyme. Thereby, the results of the present paper describing this structure are very important and, eventually, could help to develop a viable strategy to treat type 2 diabetes. These aspects make the manuscript appealing to the broad readership of Nature Communications. I support its publication. Still, in my opinion, to be compliant with the high standards of Nature Communications, some parts of the manuscript should be significantly improved:

Major point:

1. My main criticism is connected with the choice of a truncated form of UrdA. The authors used not the full-length UrdA but only its C-terminal domain. This domain binds and reduces urocanate; the structure of this very domain is important for designing specific inhibitors. Thus, this simplification is suitable for most, especially structural tasks. However, the missed N-terminal domain of UrdA is not a "putative membrane-associated domain". This additional domain contains a covalently bound FMN and serves as an electron input module of the enzyme. Without this module, UrdA-related enzymes demonstrate only residual activity, which generally does not exceed 1% of the one typical of the full-length proteins (see for example Mikoulinskaia et al. Cytochrome c-dependent methacrylate reductase from *Geobacter sulfurreducens* AM-1. *Eur J Biochem.* 1999 263: 346-352). This is also the case for UrdA. The determined activity of the C-terminal domain of UrdA is given in unusual units (pmol/min), but its conversion using the known enzyme concentration gives only 1-2 turnovers per second, which is incomparably lower than the activity of the full-length UrdA, 350 s⁻¹ (Bogachev et al., 2012). To my mind, it is doubtful to use a truncated form of UrdA in kinetic experiments, as it may significantly blur the effects of amino acid substitutions or action of inhibitors as well as pose difficulties in determining the substrate specificity. I think that all kinetic experiments should be performed only using the full-length UrdA with high catalytic activity. Production of the full-length UrdA in *E. coli* cells is naturally hampered by the absence of flavin transferase (ApbE) in

this bacterium, which is essential for covalent attachment of FMN. However, the full-length UrdA can be produced in an ApbE-containing bacterium (Bogachev et al., 2012) or by the co-expression of urdA and apbE genes in E. coli cells (Bertsova et al. Alternative pyrimidine biosynthesis protein ApbE is a flavin transferase catalyzing covalent attachment of FMN to a threonine residue in bacterial flavoproteins. J. Biol. Chem. 288, 14276-14286.).

Minor points:

1. Fig. 1d. Active site of UrdA contains an additional molecule, which is not described in the figure legend. From text at Page 17, it is possible to conclude that it is a sulfate. It is interesting to note that this sulfate occupied the position, similar to that of the E377 sidechain in fumarate reductase (Fig. S2). In fumarate reductases E377 is a part of a proton-conducting pathway to fumarate, and this residue is essential for fumarate reductase activity (Leys et al., 1999). It would be interesting to address the following question: whether sulfate being present at the active site of UrdA the consequence of high concentration of this anion during crystallization or this sulfate (or similar inorganic anion) is an essential cofactor of the enzyme. For this purpose, it would be important to determine, whether the purified UrdA contains the bound sulfate and whether the urocanate reductase activity depends on sulfate concentration in a reaction medium. This experiment should be done keeping in mind that dithionite, generally used for methyl viologen reduction, could be significantly contaminated by sulfate and bisulfite.

2.

Fig. S4. Negative values of enzymatic activities are puzzling.

Page 19, lines 373-374. Sigmoidal curve fit is unusual for monomeric enzymes.

Apparently, these strange aspects are connected with the difficulties to determine the low enzymatic activities at low substrate concentrations. At Page 19, lines 366-369 it is described that "The reductase activity was determined at each time point by subtracting absorbance from controls without substrate or enzyme, which is dependent on oxidation of the reduced methyl viologen." Possibly the conditions of measuring the reductase activity were not fully anaerobic, which resulted in a need to subtract absorbance from the controls.

If it is not so, the sigmoidal character of kinetic curves should be discussed in the text and the determined values of the Hill coefficient should be provided as well.

3. Enzymatic activities should be given in standard units ($\mu\text{mol}/\text{min}$ per mg prot. or turnovers/sec (s(-1))).

4. What does the blue curve on Fig. S4 mean?

5. Fig. 1f. Dependence of the enzymatic activities on substrate concentration is better illustrated by curves, not by bars.

6. Page 19, lines 365. Medium for the activity tests contained FAD. Does urocanate reductase activity depend on FAD concentration? If yes, the corresponding parameters should be given and discussed.

7. The C-terminal domain of UrdA is better to designate as “truncated UrdA” or UrdA', not as UrdA.

8. It would be beneficial if the authors, based on the determined 3D structure, could provide an amino acid pattern, allowing to distinguish urocanate reductases from other similar enzymes.

Alexander Bogachev

Reviewer #3 (Remarks to the Author):

This paper describes crystal structures of a fumarate reductase related enzyme, urocanate reductase and uses these to provide a model for the production of imidazole propionate. The impact of this metabolite on the glucose metabolism is interesting, but this manuscript does focus on the structure and mechanism of the enzyme rather than downstream effects of the product.

The crystallography of the individual structures appears well done, and high resolutions are obtained further giving confidence in the individual structures. The authors should provide omit electron density maps for the ligands, not model biased 2FoFc maps.

I however have some issues with the interpretation of a dynamic mechanism as outlined in Fig 3 for this enzyme. The motion of the clamp domain has been seen in fumarate reductases previously, and a similar idea is proposed here. However, the level to which individual structures are interpreted as representing distinct point on a reaction path is in my opinion debatable.

The first issue is the fact the various crystal structures are obtained in different space groups, giving distinct crystal packing constraints that can easily affect the clamp domain position. The authors should as a minimum highlight this in the text, and ideally present an analysis of how diff. packing might affect the various conformations observed. As a consequence, "intermediate" closed structures such as the Uro-FAD structure might actually be fully closed in absence of crystal packing? This would affect the interpretation of proton-donor paths etc. Can the authors model a fully closed Uro-FAD structure or are their obvious issues?

Line 119 suggest ADP binding hinders Uro binding, but inhibition by ADP merely suggests it hinders FAD binding, not necessarily Uro binding?

In addition to the crystallography, the authors present relatively limited kinetic analysis, using steady state with methyl viologen as acceptor. Values derived in terms of V_{max} and K_m should be presented with associated error values. The methods section suggest the use of various different models to fit the data from a range of mutants suggesting either poor data or unexpected complexity.

It would be much better if the authors could monitor substrate dependent-FAD reduction directly (i.e. monitor 450nm bleaching of FAD-enzyme upon addition of substrate). While this requires more enzyme, it should not be a problem if "crystallographic quantities" are routinely obtained. Binding of product to oxidised FAD might also be monitored through affects on the oxidised FAD spectrum allowing K_d determination. These experiments might also establish whether O_2 can act to recycle the $FADH_2$ back to FAD (i.e. is there (weak) oxidase activity)?

The latter point is pertinent to the paper, as given that substrate is likely reduce FAD relatively quickly, what is the evidence the Uro-FAD structure represents a kinetically competent species? One would assume this rapidly converts to Imp-FADH₂? A Uro-FADH₂ structure (i.e. colourless crystals) would be a more likely stable species, but would also assume no oxidation occurs in presence of O_2 ?

In short, while i value the individual structures and the information they provide regarding the role of various amino acids in substrate binding etc, i am not convinced of the validity of the mechanistic interpretations here unless additional analysis/kinetic data is provided.

Point-by-Point response to the comments of the reviewer.

Reviewers comments in black and our response in blue

Reviewer #1 (Remarks to the Author):

With the discovery of the multiple roles and the diversity of the microbiome in our intestine, there is literally an explosion of studies on the association between “endogenous” microbial metabolites and disease. Imidazolepropionate has been associated to type 2 diabetes and impaired glucose tolerance. This compound is generated by a reductase in the microbial pathway for histidine catabolism. The imidazolepropionate-producing urocanate reductase belongs to the well-known family of the succinate dehydrogenases/fumarate reductases. These enzymes catalyze the oxidation and/or reduction of dicarboxylic substrates and there is wealth of knowledge about them. In this manuscript, Venskutonytė et al describe their excellent crystallographic work revealing the three-dimensional structure of the urocanate reductase. High-resolution structures are described for the substrate and product complexes of the holoenzyme (FAD-bound) as well as for the enzyme in the absence of substrate/product and flavin. Mutagenesis data nicely support the conclusions drawn from the structural analysis. Urocanate reductase displays all the typical features of this enzyme family: an active-site arginine that function as proton donor and acceptor, a dynamic capping domain that opens and closes the catalytic site, a pair of Arg side chains that anchor the substrate carboxylate group, and a substrate-binding mode whereby an hydride anion can be transferred from/to the flavin. The characteristic features of urocanate reductase are mostly associated to the substrate specificity: the substrate site contains a niche for the binding of the substrate imidazole side chain. It is noticeable that the flavin itself is an H-bonding partner for the imidazole.

We thank reviewer #1 for the nice summary of our work.

Technically, the manuscript is of the highest quality. The pictures with the electron densities speak by themselves. The manuscript is instead somewhat less strong with regard to the biological insight. There is no discussion about how the structural knowledge could be used for drug inhibitor design.

We thank for this comment. Regarding the discussion on how the structure can be used for inhibitor design, we have included an additional emphasis in the discussion that the possibility to obtain high quality crystals at high resolution provides an excellent tool to study detailed interactions between the possible drug compounds and the enzyme. We believe that the availability of the structures makes the structure-based drug design open for medicinal chemists interested in this area. However, we cannot reveal specific details on the possible modifications of the compound to create inhibitor candidates, as this is a part of future patents and publications. The last paragraph now reads:

“While the extended biology of full-length UrdA is an exciting area for the future studies, in this study we examine the catalytic mechanism of the enzyme using structural

and functional data of substrate binding domains of UrdA. We demonstrate that the two-domain UrdA' is an excellent system to study ligand-protein interactions. The possibility to achieve high yields of protein and high-quality crystals allows us to obtain detailed information on bound molecules in the active site and provides a useful tool for structure-based drug design for designing inhibitors for future treatment of type 2 diabetes. In conclusion, here we report the first high-resolution structures of UrdA, an enzyme able to convert urocanate, a metabolite of L-histidine, to the non-metabolizable product imidazole propionate, which is known to interfere with human glucose metabolism. Key insights into the conformational changes associated with substrate and product binding suggest strategies for future intervention targeting the active site.”

There is very little discussion about enzyme evolution in relation to the fumarate activity reductase activity displayed by an active-site mutant. E.g. are there enzymes of the same class that operate on bulky side-chain substrates?

We agree that an additional discussion would greatly improve the manuscript. We have included a more extended discussion comparing UrdA with other oxidoreductases, including an example of a related enzyme having a bulkier substrate as well as general discussion on the biology, all included in the new section called “Discussion”.

Likewise, there is little discussion about the mechanistic implications. What is the pH optimum for the enzyme? Is there a catalytically optimal protonation state for the substrate imidazole?

Like other bacterial peptidases and proteases, UrdA is known to be active at neutral to alkaline pH (Bogachev et al., 2012; Evenepoel et al., 1999; Silvester and Cummings, 1995). At the neutral pH the imidazole ring of urocanate is expected to be partially protonated, however the physiological condition in the gut might vary in regard to pH, which can be influenced by diet and other factors. We included a paragraph on pH in the result section regarding to substrate interactions.

This now reads:

“As the uro-FAD structure was obtained at neutral pH (UrdA is active at neutral pH) the imidazole moiety of the urocanate is expected to be partially protonated. However, pH can vary in the intestine and this would affect the protonation state of the urocanate.”

Is the enzyme confirmed to be unidirectional (reference 5) or can it catalyze also imidazolepropionate oxidation?

The enzyme has been shown to be unidirectional by Bogachev et al., 2012.

How does the enzyme position itself on the membrane surface? Where is the active site entrance (Figure 2E) located with respect to the membrane surface?

In this study we have structurally examined the substrate binding domains of the UrdA. The enzyme is a lipoprotein and thus is expected to be attached to the cell membrane at the N-terminal domain, which is not present in our structures. While in this study we

show that the kinetic properties of the two-domain and the full-length UrdA are the similar, the exact positioning of the enzyme in the membrane is beyond the scope of the current study and will be investigated further in future studies.

Is the reductant (a quinone? This is not at all discussed in the manuscript) derived from the membrane? If so, how does it bind to the enzyme?

While the electron transfer in the UrdA is believed to occur via FMN in the N-terminal domain further to FAD, the external reductant is not known. However, we included a discussion on possible mechanism of the electron transfer based on comparison to other enzymes. This paragraph reads:

“In contrast to the rather well-conserved C-terminal ligand-binding domains of the oxidoreductases, the N-terminal domain varies both in sequence and function, examples being the heme-bound cytochrome domain in soluble Fcc3 and the iron-sulfur domain in quinol Frds, while yeast soluble fumarate reductase and L-aspartate oxidases lack the N-terminal domain. These differences reflect the variations in the electron transfer chain of different enzymes. UrdA has an N-terminal domain with a covalently bound FMN cofactor. FMN-containing domains are also found in NADH:quinone oxidoreductase and in the recently described extracellular fumarate reductase from Gram-positive bacteria. The latter study suggested that the N-terminal domain, where FMN also is covalently attached, directs the electron transport to the protein. The fact that the periplasmic UrdA also has such a motif might indicate a similar way of acquiring electrons, possibly from the membrane-bound quinone via cytochrome c or a different protein, but this area is open for further investigations.”

Minor point

Lines 102-107 of page 5: please clarify the text on the Y373 mutants with an explicit indication of their steady-state parameters k_{cat} and K_m .

We have reviewed the data on the Y373F and Y373H mutants in our original experiments. We have realized that the information on Y373F, which was also shown to be active doesn't add meaningful information to the manuscript and thus therefore decided to exclude this mutant from the manuscript. Regarding the Y373H mutant we have supplemented the manuscript with assays of the same mutation in the UrdA full length protein with urocanate and fumarate. Our goal was to test if Y373H mutation provides any affinity toward fumarate and the assays showed that this is indeed the case. As we are not further investigating the properties of this mutant we have decided to present the results as a comparison to wild type only, to emphasize the previously discovered finding that the residue in the position 373 does indeed play a role in distinguishing urocanate reductases from fumarate reductases (Figure S5.)

Reviewer #2 (Remarks to the Author):

The paper by Venskutonyte et al. presents the 3D structure of urocanate reductase (UrdA) from bacterium *Shewanella oneidensis*. Type 2 diabetes is associated with altered gut microbiome. The authors showed previously that this microbiome alteration results in an increase in imidazole propionate produced by UrdA-containing bacteria. Importantly, they revealed that administration of imidazole propionate in mice results in an impaired glucose tolerance (Koh et al., 2018). However, it remained unknown, whether this enrichment by UrdA-containing bacteria is the etiology or the consequence of diabetes. To answer this question, it is necessary to design specific inhibitors for UrdA, which was hampered by the lack of 3D structure of this enzyme. Thereby, the results of the present paper describing this structure are very important and, eventually, could help to develop a viable strategy to treat type 2 diabetes. These aspects make the manuscript appealing to the broad readership of Nature Communications. I support its publication.

We thank reviewer 2 for a nice summary of our work and the support for publication.

Still, in my opinion, to be compliant with the high standards of Nature Communications, some parts of the manuscript should be significantly improved:

Major point:

1. My main criticism is connected with the choice of a truncated form of UrdA. The authors used not the full-length UrdA but only its C-terminal domain. This domain binds and reduces urocanate; the structure of this very domain is important for designing specific inhibitors. Thus, this simplification is suitable for most, especially structural tasks. However, the missed N-terminal domain of UrdA is not a “putative membrane-associated domain”. This additional domain contains a covalently bound FMN and serves as an electron input module of the enzyme. Without this module, UrdA-related enzymes demonstrate only residual activity, which generally does not exceed 1% of the one typical of the full-length proteins (see for example Mikoulińska et al. Cytochrome c-dependent methacrylate reductase from *Geobacter sulfurreducens* AM-1. *Eur J Biochem.* 1999 263: 346-352). This is also the case for UrdA. The determined activity of the C-terminal domain of UrdA is given in unusual units (pmol/min), but its conversion using the known enzyme concentration gives only 1-2 turnovers per second, which is incomparably lower than the activity of the full-length UrdA, 350 s⁻¹ (Bogachev et al., 2012). To my mind, it is doubtful to use a truncated form of UrdA in kinetic experiments, as it may significantly blur the effects of amino acid substitutions or action of inhibitors as well as pose difficulties in determining the substrate specificity. I think that all kinetic experiments should be performed only using the full-length UrdA with high catalytic activity. Production of the full-length UrdA in *E. coli* cells is naturally hampered by the absence of flavin transferase (ApbE) in this bacterium, which is essential for covalent attachment of FMN. However, the full-length UrdA can be produced in an ApbE-containing bacterium (Bogachev et al., 2012) or by the co-expression of *urdA* and *apbE* genes in *E. coli* cells (Bertsova et al. Alternative pyrimidine biosynthesis protein ApbE is a flavin transferase catalyzing covalent attachment of FMN to a threonine residue in bacterial flavoproteins. *J. Biol. Chem.* 288, 14276-14286.).

We thank the reviewer for the comments. We have supplemented the study with requested experiments using full-length flavinylated UrdA. All kinetic experiments have been repeated, according to the reviewer's suggestion. The resulting K_m and k_{cat} of the full-length flavinylated UrdA are comparable to the values obtained for the two-domain construct, suggesting that the absence of N-terminal domain does not affect the enzyme activity in the assays (shown in Fig. 2b, 2c).

Minor points:

1. Fig. 1d. Active site of UrdA contains an additional molecule, which is not described in the figure legend. From text at Page 17, it is possible to conclude that it is a sulfate. It is interesting to note that this sulfate occupied the position, similar to that of the E377 sidechain in fumarate reductase (Fig. S2). In fumarate reductases E377 is a part of a proton-conducting pathway to fumarate, and this residue is essential for fumarate reductase activity (Leys et al., 1999). It would be interesting to address the following question: whether sulfate being present at the active site of UrdA the consequence of high concentration of this anion during crystallization or this sulfate (or similar inorganic anion) is an essential cofactor of the enzyme. For this purpose, it would be important to determine, whether the purified UrdA contains the bound sulfate and whether the urocanate reductase activity depends on sulfate concentration in a reaction medium. This experiment should be done keeping in mind that dithionite, generally used for methyl viologen reduction, could be significantly contaminated by sulfate and bisulfite.

We agree that this is an interesting observation. To highlight this, we have now included a discussion about the presence of the sulfate in the main text as well as the supplementary figure (Fig S11) showing an overlap between the sulfate in our structure and the E377 residue in the structure of the fumarate reductase.

This now reads:

“Interestingly, when further examining the overlaid Fcc3 and UrdA' structures, the carboxylate of Glu377 in Fcc3 superimposes exactly with a sulfate molecule found in the uro-FAD structure (Fig. S11). This sulfate seems to mimic the carboxyl group of Glu377 in Fcc3 by bridging between Arg390 and Arg411. In the imp-FAD structure, which was obtained without sulfate ions present, there is an extra density in this position that could be best accommodated by a glycerol molecule and a chloride ion. It is not clear if this has any biological relevance; rather it is likely a crystallization artefact that serves to stabilize the protein conformation in the crystal.”

However, it is not expected for the purified UrdA to contain sulfate ions as none of the buffers used for protein purification contained sulfate and any impurities are not expected to amount for high enough concentrations to saturate the protein. Moreover, imp-FAD could be crystallized with and without sulfate ions, while apo-ADP crystallized in a condition containing no sulfate, further indicating that the presence of sulfate in the uro-FAD structure is due to high concentration of this ion in the crystallization condition. While we cannot exclude the possibility of an additional binder, a possible cofactor in the binding site of UrdA, the identification of such molecule is beyond the scope of this study, as our data suggests that the sulfate in the active site area is a consequence of the specific crystallization conditions.

2. Fig. S4. Negative values of enzymatic activities are puzzling.

Page 19, lines 373-374. Sigmoidal curve fit is unusual for monomeric enzymes.

Apparently, these strange aspects are connected with the difficulties to determine the low enzymatic activities at low substrate concentrations. At Page 19, lines 366-369 it is described that “The reductase activity was determined at each time point by subtracting absorbance from controls without substrate or enzyme, which is dependent on oxidation of the reduced methyl viologen.” Possibly the conditions of measuring the reductase activity were not fully anaerobic, which resulted in a need to subtract absorbance from the controls.

If it is not so, the sigmoidal character of kinetic curves should be discussed in the text and the determined values of the Hill coefficient should be provided as well.

We agree, that the sigmoidal character of the curves requires explanation. We observe the better fit by allosteric sigmoidal equation for the wild-type enzyme with Hill coefficient of 2. We cannot explain why this is the case, but together with ITC experiments which have also indicated additional complexity in enzyme-substrate interaction we speculate that this might be caused by a shift in a monomer-dimer equilibrium of the enzyme. Interestingly the enzymatic assay data for the Y373H variant was better fitted by Michaelis-Menten equation. We think that the detailed investigation of the mode of action for the wild type enzyme as well as variants with introduced substitutions in the binding site upon urocanate binding in solution with respect to dimerization or other undiscovered allosteric events is an interesting area for further studies.

In order to address this issue, we have included a discussion in the method section describing the enzymatic assays.

3. Enzymatic activities should be given in standard units ($\mu\text{mol}/\text{min}$ per mg prot. or turnovers/sec (s^{-1})).

We have now included these parameters in figures 1a, 2b as well as in Table 2.

4. What does the blue curve on Fig. S4 mean?

The blue curve is fumarate reductase activity of Y373H mutant. However, we have now updated this figure (Figure S5) to better represent the differences between the WT and Y373H with regard to urocanate and fumarate reductase activity.

5. Fig. 1f. Dependence of the enzymatic activities on substrate concentration is better illustrated by curves, not by bars.

Since the reaction (absorbance change) was quite fast after adding reduced methyl viologen, there are variations in 0 time points depending on the experimental batches. Thus, we usually tried to include groups that we are going to compare in independent experiments. In this experiment, our purpose was to compare activities among groups and thus activity assay using different substrate concentrations were done separately. We thus

separate bar graphs to better represent our experiment settings.

However, after reviewing all the enzymatic kinetic experiments and with the addition of new data for the full length enzyme we have decided to modify this figure (now Figure S5) to only represent the finding that the Y373H mutant displays fumarate reductase activity and has a reduced urocanate reductase activity compared to the wild type enzyme.

6. Page 19, lines 365. Medium for the activity tests contained FAD. Does urocanate reductase activity depend on FAD concentration? If yes, the corresponding parameters should be given and discussed.

FAD as in the case of fumarate reductase and as shown in the structural analyses, is an important cofactor for enzymatic activity. Judged from the structures, the FAD needs to be bound for the urocanate to access the binding site. Moreover, FAD occupies a large part of the FAD-domain and is thus expected to contribute to maintaining overall protein conformational integrity. When purifying the two-domain UrdA we observe that the resulting samples are yellow (the intensity varies for different mutants/batches), indicating bound FAD already during expression in *E.coli*. However, the concentration of FAD in *E.coli* depends on expression levels and part of the protein might bind ADP (as demonstrated by our apo-ADP structure and requirement to supplement the sample with FAD to obtain the FAD bound structure). The binding is strong, as FAD does not dissociate during size exclusion chromatography. Therefore, in order to have comparable results we decided to supplement the samples for activity measurements with FAD to ensure the homogenous protein population, as the amount of FAD present in the purified sample might vary depending on the expression levels in *E.coli*.

7. The C-terminal domain of UrdA is better to designate as “truncated UrdA” or UrdA’, not as UrdA.

We agree and have changed the name for the “two-domain UrdA” to UrdA’ throughout the manuscript.

8. It would be beneficial if the authors, based on the determined 3D structure, could provide an amino acid pattern, allowing to distinguish urocanate reductases from other similar enzymes.

We thank the reviewer for this suggestion. We have examined the binding site comparing to L-aspartate oxidase, Fcc3 fumarate reductase, quinol fumarate reductase and yeast fumarate reductase and besides the already discovered difference at position 373 (histidine in other enzymes versus tyrosine in UrdA) we could distinguish A387 and F391 in UrdA, while other mentioned enzymes have glutamate and glycine in these positions respectively. We include a description and a supplementary figure (Fig S11) to illustrate these sequence differences. The new paragraph reads:

“Furthermore, when comparing UrdA with other oxidoreductases, three amino acid substitutions stand out. In addition to the tyrosine versus histidine at position 373, Phe391 of UrdA corresponds to a conserved glycine in other enzymes; and Glu377, which is part

of the proton transfer pathway in Frds, corresponds to an alanine in UrdA. Together these three residues distinguish the substrate binding site of UrdA from other related oxidoreductases (Fig. S12). “

Alexander Bogachev

Reviewer #3 (Remarks to the Author):

This paper describes crystal structures of a fumarate reductase related enzyme, urocanate reductase and uses these to provide a model for the production of imidazole propionate. The impact of this metabolite on the glucose metabolism is interesting, but this manuscript does focus on the structure and mechanism of the enzyme rather than downstream effects of the product.

We thank reviewer #3 for this accurate summary of our work.

The crystallography of the individual structures appears well done, and high resolutions are obtained further giving confidence in the individual structures. The authors should provide omit electron density maps for the ligands, not model biased 2FoFc maps.

Thanks for this comment, omit Fo-Fc maps for the ligands have been included in supplementary figure Fig S2.

I however have some issues with the interpretation of a dynamic mechanism as outlined in Fig 3 for this enzyme. The motion of the clamp domain has been seen in fumarate reductases previously, and a similar idea is proposed here. However, the level to which individual structures are interpreted as representing distinct point on a reaction path is in my opinion debatable.

The first issue is the fact the various crystal structures are obtained in different space groups, giving distinct crystal packing constraints that can easily affect the clamp domain position. The authors should as a minimum highlight this in the text, and ideally present an analysis of how diff. packing might affect the various conformations observed.

We agree with the reviewer, that crystal packing might affect the protein conformation in the crystal, however this does not seem to be an issue with our structures. We believe, that the fact that different structures crystallized in different space groups confirm the validity of the different structural arrangements, as the conformation of the clamp domain dictates the crystal packing, for example we could not obtain crystals without ligand bound in an open conformation using the same crystallization condition, furthermore the crystals in the open conformation were of lower quality indicating the effect of flexibility of the clamp domain to the crystal packing.

We have included a paragraph discussing this, reading:

“It is worth mentioning that these three states of UrdA’ showing different extents of clamp-domain closure were crystallized in different space groups with different crystal contacts. This indicates that the domain opening is not restricted by crystal packing, but instead the conformation adopted by the protein in a ligand-bound versus ligand-free state affects the crystallization. When no ligand is bound, the flexibility of the clamp domain

reduces the probability of obtaining well-diffracting crystals, as seen for apo-FAD, for which only lower quality crystals could be achieved, while the compact and closed apo-ADP and uro-FAD conformations resulted in a well-diffracting crystals.”

As a consequence, "intermediate" closed structures such as the Uro-FAD structure might actually be fully closed in absence of crystal packing? This would affect the interpretation of proton-donor paths etc. Can the authors model a fully closed Uro-FAD structure or are their obvious issues?

If we compare the apo-ADP and uro-FAD structures overlaid on the FAD domain, we can see that there would be a steric clash between the ligand and the Asp388 and Arg411 residues. Moreover, the apo-ADP structure was achieved in presence of an excess ligand in the crystallization condition, but it was still not present in the binding site in the structure, further suggesting that substrate cannot occupy the binding site in the closed conformation.

Line 119 suggest ADP binding hinders Uro binding, but inhibition by ADP merely suggests it hinders FAD binding, not necessarily Uro binding?

This is correct- the ADP competes with FAD for the same binding site. However, for the urocanate to bind, the enzyme has to first bind FAD so that the clamp domain would open, as the substrate cannot access the closed ADP bound conformation.

In addition to the crystallography, the authors present relatively limited kinetic analysis, using steady state with methyl viologen as acceptor. Values derived in terms of V_{max} and K_m should be presented with associated error values.

We have now included a table (Table 1) with kinetic parameters including the error values.

The methods section suggest the use of various different models to fit the data from a range of mutants suggesting either poor data or unexpected complexity.

As noted by the Reviewer, we believe that this is due to the unexpected complexity, which was also observed in our ITC data. One of the explanations for this complexity might be the shift between the monomer and dimer equilibrium of the UrdA in solution.

It would be much better if the authors could monitor substrate dependent-FAD reduction directly (i.e. monitor 450nm bleaching of FAD-enzyme upon addition of substrate). While this requires more enzyme, it should not be a problem if "crystallographic quantities" are routinely obtained. Binding of product to oxidised FAD might also be monitored through affects on the oxidised FAD spectrum allowing K_d determination. These experiments might also establish whether O_2 can act to recycle the FADH₂ back to FAD (i.e. is there (weak) oxidase activity)?

We agree that determining the K_d would be a valuable addition to the study. We have

performed ITC studies, which in addition to K_d provides also other parameters of ligand binding and contributes to further understanding of the enzyme interactions with the substrate (shown in Fig. 2d, Fig. S6 and Table 2).

The latter point is pertinent to the paper, as given that substrate is likely reduce FAD relatively quickly, what is the evidence the Uro-FAD structure represents a kinetically competent species?

The structure represents the binding mode of the substrate and the FAD. The oxidation state of FAD in the structure cannot be known for sure, but the reaction is not expected to occur in the crystal due to the lack of electron donor/anaerobic conditions. The redox state of FAD is dependent on various factors, we have included the discussion on that in the method section. The main goal of the study was to establish the ligand-protein interactions as well as accompanying conformational changes, which the structures clearly illustrate. We agree that determining the redox state of the FAD in the crystal would be very interesting, however that is beyond the scope of our study.

One would assume this rapidly converts to Imp-FADH₂? A Uro-FADH₂ structure (i.e. colorless crystals) would be a more likely stable species, but would also assume no oxidation occurs in presence of O₂?

We do not know if upon data collection the crystal represents FAD, FADH or FADH₂. The yellow color of the crystals indicates the oxidized form of the FAD, but there are studies showing that in most cases the FAD will be partially or fully reduced. This is not very relevant in terms of the crystal structure, which is achieved in excess of both substrate and FAD and with no expected catalytic activity (due to lack of electron donor). To determine intermediate states of the enzymatic activity including the reduction of FAD and further reduction of the substrate would require time resolved studies.

In short, while i value the individual structures and the information they provide regarding the role of various amino acids in substrate binding etc, i am not convinced of the validity of the mechanistic interpretations here unless additional analysis/kinetic data is provided.

We have provided additional kinetic data (ITC) to validate the mechanistic interpretations.

References

- Bogachev, A.V., Bertsova, Y.V., Bloch, D.A., and Verkhovsky, M.I. (2012). Urocanate reductase: identification of a novel anaerobic respiratory pathway in *Shewanella oneidensis* MR-1. *Molecular microbiology* 86, 1452-1463.
- Evenepoel, P., Claus, D., Geypens, B., Hiele, M., Geboes, K., Rutgeerts, P., and Ghoo, Y. (1999). Amount and fate of egg protein escaping assimilation in the small intestine of humans. *Am J Physiol-Gastr L* 277, G935-G943.

Karlsson, F.H., Tremaroli, V., Nookaew, I., Bergstrom, G., Behre, C.J., Fagerberg, B., Nielsen, J., and Backhed, F. (2013). Gut metagenome in European women with normal, impaired and diabetic glucose control. *Nature* 498, 99-+.

Silvester, K.R., and Cummings, J.H. (1995). Does Digestibility of Meat Protein Help Explain Large-Bowel Cancer Risk. *Nutr Cancer* 24, 279-288.

REVIEWERS' COMMENTS

Reviewer #1 (Remarks to the Author):

The Authors have carefully and convincingly addressed the points made by the Reviewers. This includes the critical issue about the full-length protein (Reviewer 2). This is very good manuscript.

Andrea Mattevi

Reviewer #2 (Remarks to the Author):

The revised version of the manuscript by Venskutonyte et al. has been significantly improved. All my previous concerns have been addressed, and the manuscript can be published in Nature Communications.

Alexander Bogachev

Reviewer #3 (Remarks to the Author):

This manuscript is now improved and has addressed many of the reviewers comments.

I support publication, providing the authors report K_m and V_{max} values accurately, rather than overly precise. In other words, associated error should be reported to at most have two significant numbers, and the K_m V_{max} value should be reported with corresponding precision.